# Analysis of Risky Riding Behavior Characteristics of the Related Road Traffic Injuries of Electric Bicycle Riders

**DOI:** 10.3390/ijerph20075352

**Published:** 2023-03-31

**Authors:** Jiayu Huang, Ziyi Song, Linlin Xie, Zeting Lin, Liping Li

**Affiliations:** 1School of Public Health, Shantou University, Shantou 515041, China; 22jyhuang1@stu.edu.cn (J.H.); 22zysong@stu.edu.cn (Z.S.); 22llxie@stu.edu.cn (L.X.); 18ztlin@stu.edu.cn (Z.L.); 2Injury Prevention Research Center, Shantou University Medical College, Shantou 515041, China

**Keywords:** road traffic injury, electric bicycle, riding behavior, epidemiological characteristics, descriptive study, roadside observation

## Abstract

Electric bicycle (EB) riders, being vulnerable road users (VRUs), are increasingly becoming victims of road traffic injuries (RTIs). This study aimed to determine the current status and epidemiological characteristics of RTIs among EB riders through a questionnaire survey and roadside observations in Shantou to provide a scientific basis for the prevention and control of electric bicycle road traffic injuries (ERTIs). A total of 2412 EB riders were surveyed, and 34,554 cyclists were observed in the study. To analyze the relationship between riding habits and injuries among EB riders, chi-square tests and multi-factor logistic regression models were employed. The findings reveal that the prevalence of ERTIs in Shantou was 4.81%, and the most affected group was children under 16 years old, accounting for 9.84%. Risky behavior was widespread among EB riders, such as the infrequent wearing of safety helmets, carrying people on EBs, riding on sidewalks, and listening to music with headphones while bicycling. Notably, over 90% of those who wore headphones while bicycling engaged in this risky behavior. The logistic regression analysis showed that honking the horn (odds ratio (OR): 2.009, 95% CI: 1.245–3.240), riding in reverse (OR: 4.210, 95% CI: 2.631–6.737), and continuing to ride after a fault was detected (OR: 2.010, 95% CI: 1.188–3.402) all significantly increased the risk of ERTIs (all *p* < 0.05). Risky riding behavior was significantly less observed at traffic intersections with traffic officers than at those without (all *p* < 0.001).

## 1. Introduction

Road traffic injuries (RTIs) are a major public health concern globally, with the number of deaths continuing to increase each year. In 2018 and 2019, the World Health Organization (WHO) reported that 1,390,000 and 1,350,000 people, respectively, died in traffic accidents worldwide [1,2]. Apart from the loss of lives, RTIs also have significant economic implications, accounting for about 3% of a country’s gross domestic product (GDP) [3]. The situation is particularly alarming in low- and middle-income countries, where the burden of road traffic injuries and deaths is disproportionately high (90%) due to rapid urbanization and motorization [4,5]. Although China, the world’s largest developing country, has experienced a decrease in the frequency of RTIs following road expansion, the number of fatalities in non-motorized traffic accidents has continued to rise since 2012, leading to an increasing economic burden [6,7]. In 2018, there were about 2,162,000 road traffic accidents in China, resulting in 63,957 deaths and 386,000 and 396,000 cases of serious and minor injuries, respectively. The direct economic loss caused by these accidents was RMB 243.63 billion, as reported in the 2019 Annual Report on Road Traffic Safety in China by the Institute of Highway Science of the Ministry of Transport of China. According to the global burden of disease research data, China accounted for 21% of global road traffic fatalities in 2017, equivalent to 0.6% of GDP [8].

Electric bicycle (EB) is a two-wheeled vehicle that is powered by an onboard battery as an auxiliary energy source. Due to their ease of use, cost-effectiveness, and convenience, EBs are gaining popularity worldwide as the ideal mode of transportation for modern society. Private electric bicycles (PEBs) and shared electric bicycles (SEBs) are commonly used by riders for daily travel. However, as vulnerable road users (VRUs), EB cyclists are increasingly becoming involved in road traffic injuries (RTIs). For instance, in Switzerland, the number of injuries reported by the police while using e-bikes tripled between 2011 and 2018, reaching nearly 700 in 2016 [9]. Between 2000 and 2017, the U.S. National Electronic Injury Surveillance System recorded 133,872 injuries related to e-bikes and e-scooters, which accounted for 5.3% of all emergency department injuries, with 17% of EBs causing serious injuries such as cranial trauma and internal bleeding [10]. From 2014 to 2020, the number and severity of e-bike injuries seen in emergency departments dramatically increased [11]. In China, ERTIs have become a significant problem, with accidents, injuries, and economic losses increasing as a proportion of RTIs [12,13,14]. For instance, in 2008, 5.4% of all traffic fatalities in China were caused by e-bike riders, which increased to 7.8% in 2012 and 12.0% in 2016 [15]. ERTIs have become a road safety issue that requires urgent attention.

In recent years, more studies have focused on VRUs, including e-bike riders. For instance, Keila González-Gómez’s study identified poor visibility and insufficient sight distance as factors contributing to traffic accidents involving VRUs, while Dasanayaka et al. identified driver error and VRU error as key factors in vehicle-VRU collisions and reduced visibility and slower reaction time as key factors contributing to collisions [16,17]. Scarano et al. analyzed the cyclist safety research of nearly 10 years by bibliometric methods and found that e-bike and behavior are the main keywords, which showed that people pay special attention to e-bikes and riding behaviors [18]. A review of risky riding behaviors of urban e-bike riders showed that illegal occupancy of motor vehicle lanes, speeding, red light running, illegal carrying, and riding in the opposite direction were the main risky riding behaviors of e-bikes, which increase the risk of ERTIs [19]. Similarly, a case-control study conducted in Hefei, China, illustrated that the risky riding behaviors of cyclists violating traffic rules are important influencing factors of ERTIs [20]. Furthermore, using cell phones while cycling within 1 min of a road injury was linked to a three-fold increase in the probability of injury [21]. It has been demonstrated that e-bike users have a lower perception rate than users of other road transport vehicles, which is one of the reasons for the dramatic increase in the prevalence of ERTIs in recent years [22]. In addition, age and gender are also key influences on ERTIs. A retrospective study on spinal injuries in a Chinese city involving dynamic bicycle-related collisions demonstrated that victims of bicycle-related collisions resulting in spinal injuries were mostly male (79.07%) and middle-aged (41–60 years, 44.19%) [23]. Another study investigating e-bike collisions and their effect on orthopedic injuries also indicated that male patients were more likely than female patients to suffer multiple fractures and related injuries [24].

In summary, the causes of ERTIs are multifaceted and complex. Current studies on RTIs mainly focus on cars and bicycles, and there is a lack of research on the characteristics of cycling behaviors and the occurrence of ERTIs among EB cyclists in small and medium-sized cities in China. This study employs a cross-sectional survey and roadside observation to analyze and uncover the awareness of EB road safety regulations, the cycling behavioral characteristics, and the current situation of road traffic injuries among EB riders in Shantou. We will first provide the basic information about e-bike riders and the basic situation of ERTIs in Shantou City, then use a multifactorial unconditional logistic regression model to analyze these data and, finally, obtain the influencing factors of ERTIs. The goal of this study is to provide a scientific basis for the prevention and reduction of ERTIs by elucidating their epidemiological characteristics. It is of practical significance to ensure people’s safe travel and improve the public transportation system.

## 2. Methods

### 2.1. Cross-Sectional Study

#### 2.1.1. Participants and Procedure

Between January and June 2021, a multi-stage stratified whole-group random sampling method was employed to conduct a questionnaire survey in Shantou City. Since Shantou is a small and medium-sized city, parents of young children and elementary school students mostly rely on EBs for transportation, while other students with independent mobility tend to use EBs instead of walking. Therefore, we chose the school as the study area. Schools were randomly selected in each of the urban and suburban areas, including kindergarten, elementary school, middle school, high school, and university. A total of nine schools were selected because there were no universities in the suburbs. Students and parents who met the inclusion and exclusion criteria were randomly selected as respondents in these schools. The inclusion criteria were having at least three EB rides (including PEBs or SEBs) in Shantou City, each lasting at least five minutes, while the exclusion criteria were not receiving informed consent or completing the questionnaire invalidly. In this study, ERTIs were defined as non-fatal injuries caused by riding, pushing, or riding an EB in the past 12 months where the rider presented to a medical facility and was diagnosed with a specific injury or took more than one day off work because of the injury. Due to the COVID-19 outbreak, electronic and offline paper questionnaires were used. The questionnaire was co-developed by a team of researchers from the Injury Prevention Research Center, refined and optimized through modifications and pre-surveys by injury epidemiology research experts, and had good reliability indicated by a Cronbach’s alpha value of 0.781. It covered the study population’s basic demographic characteristics, awareness of road safety regulations, characteristics of riding behavior, and the occurrence of ERTIs.

#### 2.1.2. Sample Size Calculation

The sample size was calculated using the formula (*n* = (Z_α_)^2^ *p* (1 − *p*)/d^2^); whereas *n* = sample size, Z_α_ (1.96): significance level at α = 0.05, *p*: expected proportion (10%), d: margin of error = 0.15 *p* (0.015). Considering a 30% increase in sample size for invalid responses, the sample size for this study should not be less than 2083 participants to be statistically acceptable. Therefore, a total of 2444 questionnaires were distributed, and 2412 valid questionnaires were returned, for an effective rate of 98.69%.

### 2.2. Roadside Observational Study

In March 2021, six intersections in downtown Shantou with heavy traffic, high pedestrian flow, and clear visibility and traffic signals were selected as observation areas, while the suburbs of Shantou were not chosen because they do not have bicycle sharing. These sites were observed during the peak commuting hours of 8:00 a.m. to 9:00 a.m.; 11:30 a.m. to 12:30 p.m.; 2:00 to 3:00 p.m.; and 5:30 to 6:30 p.m. The observations included the gender of the EB user (male or female), the type of EB (private or shared), and six types of riding behavior: (1) whether wearing a helmet; (2) whether carrying a passenger; (3) whether riding on the sidewalk; (4) whether riding against the light; (5) whether running a red light; and (6) whether holding a cell phone. All observers were trained in advance, and they went to the observation point for a pre-observation to familiarize themselves with the observation position and observation methods. After each observation, the observation record sheets were checked and verified by two people to ensure the completeness and accuracy of the observations.

### 2.3. Data Analysis

The data were double entered and consistency checked with EpiData 3.2 and were statistically analyzed with SPSS 26.0 (IBM, Armonk, NY, USA). Mean ± standard deviation and frequency (composition ratio) were used for the statistical description of measures and counts, respectively. The cardinality test was used to compare the incidence of ERTIs between groups. A multifactorial unconditional logistic regression model was developed to analyze the risk factors of ERTIs. The odds ratio (OR) and its 95% confidence interval (95% CI) were estimated. The test level of α = 0.05 was set, and all were two-sided tests.

## 3. Results

### 3.1. Cross-Sectional Study

#### 3.1.1. Demographic Characteristics of EB Riders

A total of 2412 EB riders were validly surveyed, of whom 116 reported having suffered an ERTI in the past 12 months, with an ERTI prevalence rate of 4.81% (116/2412). The prevalence of injury was higher among the males (7.62%) than the females (3.37%), and the students (7.00%) than the parents (2.37%). Children under 16 years old had a high incidence of injury, with a statistically significant difference of approximately eight percentage points higher than other age groups (9.84%) (*p* < 0.001). The injury prevalence among cyclists with different educational levels was significantly different. However, there was no statistically significant difference in injury prevalence among cyclists in different survey regions (*p* > 0.05), as shown in Table 1.

#### 3.1.2. EB Riders’ Knowledge of Road Safety Regulations

Table 2 shows the cognitive items in ascending order of EB road safety regulation awareness. There was a significant difference between ERTI cyclists and non-ERTI cyclists in their awareness of related cycling norms such as that EB riders must be at least 16 years old, EB riders should not ride with an umbrella on a rainy day, EB riders should not ride in the opposite direction, EB riders should not ride after drinking, EB riders should wear a safety helmet, and EBs should not be ridden on the motorway (*p* < 0.05).

#### 3.1.3. Riding Behavior Characteristics of EB Riders

Compared to factors that are difficult to artificially overcome, such as wheel slippage (2.82%), brake failure (2.03%), and feeling nervous (2.94%), the three major risky riding behaviors were less frequent for regular drunk riding (1.29%), riding in the opposite direction (1.33%), and disobeying traffic signals (1.24%). However, the dangerous riding behaviors of infrequently wearing helmets (41.96%), carrying people on bicycles (68.78%), riding on sidewalks (52.32%), and listening to music while riding with headphones on (93.66%) were more common, with the frequency of wearing headphones and playing music while riding being over 90%. The proportion of those who regularly have safe riding behaviors is less than 60%. Except for placing or hanging heavy objects in the basket or on the handlebars, all other risky riding behaviors significantly increased the incidence of ERTIs (*p* < 0.001). The incidence of ERTIs was higher in those with frequent risky cycling behaviors than in those with occasional risky cycling behaviors, as shown in Table 3.

#### 3.1.4. Multifactorial Logistic Analysis of Risk Factors for ERTIs

Females had a lower risk of injury than males (OR: 0.49, 95% CI: 0.323–0.743, *p* < 0.001). The risk of injury showed varying degrees of reduction with increasing age, with children under 16 years of age being at high risk of injury, while those over 36 years of age were at relatively low risk (OR: 0.16, 95% CI: 0.068–0.350, *p* < 0.001). Horn honking, riding in the opposite direction, and continuing to ride after a malfunction was detected all contributed to a significantly increased risk of injury. The behavior of sounding the horn and continuing to ride after a malfunction was about two times higher than the risk of sounding the horn when never riding, while the risk of injury occurrence was 4.21 times higher for those with reverse riding behavior than those who did not (95% CI: 2.631–7.737, *p* < 0.05), as shown in Table 4.

### 3.2. Roadside Observation

A total of 34,554 cyclists were observed on the roadside, of which 14,144 (40.93%) rode PEBs and 20,410 (59.07%) rode SEBs. A total of 17,433 (50.45%) were male and 17,121 (49.55%) were female. PEB riders were significantly more likely than SEB riders to engage in risky riding behaviors such as riding in traffic, running red lights, riding on sidewalks, answering phones, and reading cell phone messages while riding. The occurrence of these dangerous riding behaviors was significantly higher at traffic intersections without traffic policing than at intersections with traffic policing (*p* < 0.001). The differences in the distribution of the different periods were statistically significant (*p* < 0.001), as shown in Table 5.

## 4. Discussion

The results of this study showed that the prevalence of ERTIs in Shantou was 4.81%. This is similar to the results of a US study that showed ERTIs accounted for 5.3% of all emergency department visits between 2000 and 2017 [10]. However, the results of this study were lower than the findings from Yixing City, Jiangsu Province, southern China, in 2015 (15.99%) [25]. This may be due to the fact that the population participating in this survey is a group of students and parents, while the main users of EBs are not only students and parents who pick up their children but also more unmarried and young working people. Therefore, the sample of this study may be underestimated compared to the community residents of Yixing City.

By comparing the incidence of ERTIs among different demographic characteristics of cyclists, it is clear that there are significant differences in the incidence of ERTIs by gender, age, and educational level. The prevalence of ERTIs is significantly higher in men than in women, which may be due to the fact that men ride faster, prefer risky behavior while riding, and are more likely to commit road safety violations compared to women [9,26]. In addition, there were differences in the distribution of ERTIs in different age groups. The results of this study showed that the incidence of ERTIs was highest among people under the legal riding age (16 years old), especially for junior high school students. This may be due to the low literacy level of junior high school students and the lack of guidance and supervision for them. Wang et al. found that people with low literacy levels are prone to operational errors during cycling, and factors such as a relatively weak awareness of self-protection, a lack of traffic safety knowledge, and a lessened understanding of traffic regulations make them more likely to be vulnerable to traffic accidents [27]. The results of the study, which investigated the knowledge of EB-related road safety regulations among school students and parents, showed that EB riders’ knowledge of the law was generally low, with only 41.25% of them correctly knowing that they had to be at least 16 years old to ride an electric bicycle. The lack of knowledge about safe riding among EB riders may be the reason for the frequent occurrence of various dangerous riding behaviors [22,28].

The results of the questionnaire survey on cyclists’ cycling behavior characteristics in this study showed that the longer the one-way distance traveled and the longer the average daily cycling hours, the significantly higher the incidence of ERTIs. More than 90% of e-cyclists wear earphones while riding. A study in Germany also found that it was common for riders to wear headphones while riding [29]. This requires attention because the act of wearing headphones and listening to music may narrow the cyclist’s breadth of attention, reducing his or her ability to obtain complete information about road conditions and thus failing to pay attention to potential hazards, thus missing the best time to avoid the risk of injury [30]. In the United States, seven states have explicitly banned the use of headphones by cyclists, but there are no clear rules on the matter in China. The results of the single-factor analysis showed that in addition to placing or hanging heavy objects in the basket or on the handlebars, emergency braking, sounding the horn, riding after drinking alcohol, never wearing a safety helmet while riding, riding with a person on the bike, riding in the opposite direction, not obeying traffic signals, riding with one hand, wearing headphones and playing music while riding, chasing and jostling with other vehicles or pedestrians, using a cell phone while riding, performing stunts while riding, continuing to ride after a malfunction, riding on the motorway, riding on the sidewalk, and riding nearsighted but without glasses are all dangerous riding behaviors significantly associated with the occurrence of ERTIs. EBs, as non-motorized vehicles, do not travel as fast as motor vehicles, and the relevant laws and regulations are still inadequate. Therefore, cyclists are prone to take a chance and engage in dangerous riding behaviors when riding. Risky riding behaviors, such as riding after drinking alcohol [31,32,33], riding without a safety helmet [34,35,36,37], riding with a passenger [25,35], riding in the opposite direction [25,36], not obeying traffic signals [25,33,38], using a cell phone or wearing headphones while riding [36,39,40], chasing and jostling with other vehicles or pedestrians [36], continuing to ride after discovering a vehicle malfunction [36], and riding on motor vehicle lanes [33,41] have been confirmed to be associated with the occurrence of ERTIs, consistent with the results of this study.

The results of the multifactorial logistic regression in this study showed that horn honking, riding in the opposite direction, and continuing to ride after a fault was detected all significantly increased the risk of ERTIs. Among them, the risk of injury occurrence was 4.21 times higher for those who had reverse riding behavior than for those who did not. Riding in the opposite direction is a very serious road traffic violation, which not only disrupts the road traffic order but also further increases the risk of collision with other road traffic participants. Since e-bikes are relatively smaller and more convenient to use than cars, the operation of retrograde is easier, and the road traffic department also lacks electronic monitoring and punishment policies for such vehicles, resulting in the prevalence of retrograde behavior among e-bike riders and the increased incidence of RTIs.

The roadside observation study found that PEB riders showed significantly higher rates of risky riding behaviors such as going against traffic, running red lights, riding on sidewalks, and reading cell phone messages while riding than SEB riders, which may be due to the poor supervision of PEBs by traffic police in Shantou City. The study also found that the above-mentioned risky riding behaviors were significantly more frequent at traffic intersections without traffic police officers than those with traffic police officers. In terms of the distribution of different peak commute hours, riding on the sidewalk and reading cell phone messages while riding was predominant in the period of 11:30 p.m.–12:30 p.m., and the dangerous riding behaviors of going against the traffic, running red lights, and answering phone calls while riding were all high in the afternoon from 17:30 p.m. to 18:30 p.m.

In light of the current prevalence of ERTIs, it is crucial to implement measures for their prevention and control. Initially, the focus should be on junior high school students since adolescence is a crucial stage for developing good habits. Therefore, enhancing the safety of students’ travel education and instilling good travel habits from an early age is essential. To achieve this, the Traffic Bureau of the Ministry of Public Security can collaborate with the Ministry of Education and the Communist Youth League to conduct road safety education and awareness campaigns through relevant presentations. Additionally, dangerous riding behavior must be regulated by formulating and revising road safety laws that are strictly implemented [42,43]. This can effectively decrease the incidence and mortality rate of RTI, especially in low- and middle-income countries [5]. Targeted supervision by traffic police during high-risk periods can also prevent dangerous riding behavior. Finally, to prevent ERTIs, restrictions on carrying behavior and mandatory helmet wearing for SEBs should be emphasized. Governments must establish effective cooperation with SEB companies to address these issues.

This study utilized both descriptive and roadside observational approaches to determine the prevalence and characteristics of ERTIs in Shantou, China, with the aim of identifying high-risk groups for ERTIs and the corresponding risky riding behaviors that require attention. However, the study is subject to certain limitations that should be acknowledged. Firstly, it is a cross-sectional study that can only establish associations between ERTIs and risk factors but cannot infer causation. Secondly, recall bias may have occurred due to the self-administration of the questionnaire by investigators in the cross-sectional study, leading to a potential underestimation of results. Thirdly, the roadside observation study may have been limited by the manual counting of cyclists’ riding behaviors by observers, potentially leading to omissions in the data. Due to safety concerns for the observers, we did not investigate the roundabouts, even though they are considered intersections that are difficult to navigate for cyclists and have a significant impact on their safety [44]. Future studies at roundabouts could be possible through video or other formats. Additionally, logistic regression analysis was employed in the study, but it is unsuitable for solving nonlinear problems and heavily relies on accurate data representation. Future studies could incorporate alternative statistical analysis methods, such as decision trees and K-nearest neighbor, to enhance the analytical value of the research. What is more, we focused on cyclists’ safety perceptions and riding behaviors while neglecting some factors that may influence those behaviors, such as traffic conditions and weather, which can be further explored in the future. Finally, this study was conducted solely in Shantou, a small-to-medium-sized city located in the coastal monsoon area of southeast Guangdong. Therefore, the findings cannot be generalized to the entire country or the entire world.

## 5. Conclusions

The current situation regarding risky cycling behaviors and ERTIs in Shantou, a southern city in China, is worrisome. The awareness of the law among EB riders is generally low, and there are numerous instances of risky riding behaviors. These behaviors include riding without safety helmets, carrying passengers on bikes, cycling on sidewalks, and listening to music with headphones on. Other risky behaviors such as honking the horn, riding in the opposite direction, and continuing to ride after detecting a fault significantly increase the risk of ERTIs. This study highlights the importance of targeting middle school students under 16 years of age in reducing the occurrence of risky riding behaviors through enhanced safety education and more effective management systems. It is recommended that relevant governments and traffic safety departments establish a good working relationship with schools and SEB enterprises to provide education and training on road safety regulations and awareness. This will help to improve their knowledge of traffic violations and safe riding behaviors and prevent the occurrence of ERTIs.

## Figures and Tables

**Table 1 ijerph-20-05352-t001:** ERTIs of cyclists with different demographic characteristics, *n* (%).

Variables	Total (*n* = 2412)	Injured Cyclists (*n* = 116)	$\chi^{2}$	*p*-Value
Gender			27.777	<0.001
Male	853 (35.36)	65 (65.03)		
Female	1559 (64.64)	51 (43.97)		
Age			63.790	<0.001
11~	762 (31.59)	75 (64.66)		
16~25	523 (21.68)	12 (10.34)		
26~35	656 (27.20)	22 (18.97)		
36~	471 (19.53)	7 (6.03)		
Participant			28.227	<0.001
Parent	1141 (47.31)	27 (23.28)		
Student	1271 (52.69)	89 (76.72)		
Investigated Area			2.318	0.128
Urban	1452 (60.20)	62 (53.45)		
Suburb	960 (39.80)	54 (46.55)		
Educational Level			34.710	<0.001
Elementary or below	126 (5.22)	7 (6.03)		
Middle school	931 (38.60)	73 (62.93)		
High school	745 (30.89)	24 (20.69)		
College or bachelor’s	542 (22.47)	9 (7.76)		
Master’s or above	68 (2.82)	3 (2.59)		

**Table 2 ijerph-20-05352-t002:** Awareness of EB road safety regulations, *n* (%).

Items	The Correct	The Injured	$\chi^{2}$	*p*-Value
The speed of EBs should be lower than 25 km/h.	495 (20.52)	19 (3.84)	8.061	0.089
EB riders should wear a safety helmet.	513 (21.26)	35 (6.82)	5.769	0.016
Adult EB riders can only carry a child under the age of 12.	915 (37.93)	35 (3.83)	6.954	0.138
EB riders must be at least 16 years old.	995 (41.25)	47 (4.72)	11.870	0.018
EB riders should not use their cell phones while riding.	1942 (80.51)	86 (4.43)	3.158	0.076
EB riders should not ride with an umbrella on a rainy day.	2230 (92.45)	97 (4.35)	13.631	<0.001
EBs should not be ridden on the sidewalk.	2249 (93.24)	103 (4.58)	3.828	0.050
EB riders should not ride in the opposite direction.	2250 (93.28)	99 (4.40)	20.790	<0.001
EB riders should not ride after drinking.	2273 (94.23)	103 (4.53)	6.650	0.010
EBs should not be ridden on the motorway.	2305 (95.56)	106 (4.60)	5.033	0.025

**Table 3 ijerph-20-05352-t003:** Riding behavior characteristics of EB riders, *n* (%).

Riding Behavior	Total	Injured Cyclists	$\chi^{2}$	*p*-Value
Emergency braking			28.249	<0.001
Never	684 (28.36)	20 (17.24)		
Sometimes	1505 (62.40)	70 (60.34)		
Always	223 (9.25)	26 (22.41)		
Honking the horn while riding			66.086	<0.001
Never	1150 (47.68)	26 (22.41)		
Sometimes	1087 (45.07)	62 (53.45)		
Always	175 (7.26)	28 (24.14)		
Riding after drinking alcohol			35.608	<0.001
Never	2210 (91.63)	94 (81.03)		
Sometimes	171 (7.09)	14 (14.07)		
Always	31 (1.29)	8 (6.90)		
Wearing a safety helmet while riding			16.946	<0.001
Never	404 (16.75)	16 (13.79)		
Sometimes	608 (25.21)	48 (41.38)		
Always	1400 (58.04)	52 (44.83)		
Hanging or placing heavy objects			4.451	0.108
Never	1028 (42.62)	39 (33.62)		
Sometimes	1233 (51.12)	67 (57.76)		
Always	151 (6.26)	10 (8.62)		
Carrying people on bicycles			17.265	<0.001
Never	753 (31.22)	32 (27.59)		
Sometimes	1185 (49.13)	44 (37.93)		
Always	474 (19.65)	40 (34.48)		
Retrograde			104.454	<0.001
Never	1713 (71.02)	45 (38.79)		
Sometimes	667 (27.65)	60 (51.72)		
Always	32 (1.33)	11 (9.48)		
Failure to obey traffic signals			53.281	<0.001
Never	1980 (82.09)	69 (59.48)		
Sometimes	402 (16.67)	40 (34.48)		
Always	30 (1.24)	7 (6.03)		
One hand off the handlebars			28.363	<0.001
Never	1713 (71.02)	63 (54.31)		
Sometimes	639 (26.49)	43 (37.07)		
Always	60 (2.49)	10 (8.62)		
Wear headphones and play music while riding			49.301	<0.001
Never	153 (6.34)	16 (13.79)		
Sometimes	2018 (83.67)	70 (60.34)		
Always	241 (9.99)	30 (25.86)		
Chasing and jostling with other vehicles or pedestrians			43.319	<0.001
Never	2109 (87.44)	79 (68.10)		
Sometimes	223 (9.25)	25 (21.55)		
Always	80 (3.32)	12 (10.34)		
Phone calls while riding			31.089	<0.001
Never	1728 (71.64)	58 (50.00)		
Sometimes	583 (24.17)	46 (39.66)		
Always	101 (4.19)	12 (10.34)		
Performing stunts such as rocking cars			11.991	0.002
Never	1857 (76.99)	74 (63.79)		
Sometimes	347 (14.39)	26 (22.41)		
Always	208 (8.62)	16 (13.79)		
Discovered vehicle failure and continued riding			68.137	<0.001
Never	2165 (89.76)	78 (67.24)		
Sometimes	159 (6.59)	26 (22.41)		
Always	88 (3.65)	12 (10.34)		
Riding on the motorway			65.812	<0.001
Never	1305 (54.10)	41 (35.34)		
Sometimes	913 (37.85)	43 (37.07)		
Always	194 (8.04)	32 (27.59)		
Riding on the sidewalk			28.633	<0.001
Never	1150 (47.66)	39 (33.62)		
Sometimes	1077 (44.65)	54 (46.55)		
Always	185 (7.67)	23 (19.83)		
Myopia but riding without glasses			50.812	<0.001
Never	1939 (80.39)	65 (56.03)		
Sometimes	397 (16.46)	39 (33.62)		
Always	76 (3.15)	12 (10.34)		

**Table 4 ijerph-20-05352-t004:** Multifactorial logistic regression analysis of risk factors for ERTIs.

Risk Factors	β	$\chi^{2}\;$	OR	OR 95%CI	*p*-Value
Gender					
Male			1000		
Female	−0.714	11.259	0.490	0.323–0.743	<0.001
Age					
11~			1.000		
16~25	−1475	20.714	0.229	0.121–0.432	<0.001
26~35	−0.965	11.861	0.381	0.220–0.660	0.001
36~	−865	20.053	0.155	0.068–0.350	<0.001
Honking the horn while riding	0.697	8.177	2.009	1.245–3.240	0.004
Retrograde	1.437	35.912	4.210	2.631–6.737	<0.001
Discovered vehicle failure and continued riding	0.698	6.765	2.010	1.188–3.402	0.009

**Table 5 ijerph-20-05352-t005:** Comparison of dangerous riding behavior, *n* (%).

Variable	Number	Retrograde	Ignoring Signal	Sidewalk Riding	Answering Phone	Reading Phone
Types of EB						
PEB	14,144	698 (4.93)	872 (6.17)	4417 (31.23)	528 (3.73)	1212 (8.57)
SEB	20,410	865 (4.24)	1055 (5.17)	5260 (25.77)	637 (3.12)	1416 (6.94)
$\chi^{2}$		9.393	15.743	123.396	9.605	31.635
*p*-Value		0.002	<0.001	<0.001	0.002	<0.001
Traffic Police						
Yes	8394	147 (1.75)	122 (1.45)	1941 (23.12)	122 (1.45)	484 (5.77)
No	25,942	1416 (5.46)	1805 (6.96)	7736 (29.82)	1043 (4.02)	2144 (8.26)
$\chi^{2}$		200.591	362.741	140.517	127.503	56.016
*p*-Value		<0.001	<0.001	<0.001	<0.001	<0.001
Period						
8:00–9:00	9473	350 (3.69)	510 (5.38)	2844 (30.02)	164 (1.73)	494 (5.21)
11:30–12:30	7873	378 (4.80)	317 (4.03)	2289 (29.07)	360 (4.57)	888 (11.28)
14:00–15:00	7158	276 (3.86)	445 (6.22)	1715 (23.96)	191 (2.67)	446 (6.23)
17:30–18:30	9832	559 (5.69)	655 (6.66)	2829 (28.77)	450 (4.58)	800 (8.14)
$\chi^{2}$		54.209	63.700	83.706	166.737	249.873
*p*-Value		<0.001	<0.001	<0.001	<0.001	<0.001
Total	34,554	1563 (4.52)	1927 (5.58)	9677 (28.01)	1165 (3.37)	2628 (7.61)

## Data Availability

The datasets used and analyzed in the study are available from the corresponding author upon reasonable request.

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
