# Peer review of "Analysis of Risky Riding Behavior Characteristics of the Related Road Traffic Injuries of Electric Bicycle Riders"

_ijerph, 2023, doi:10.3390/ijerph20075352_

Round 1

Reviewer 1 Report

The current paper has proposed an analysis of risk riding behavior and related traffic injuries among electric bicycle riders. The subject presented is interesting and overall the paper is written well, however, there are a few points/queries to be addressed and clarified. Please see the comments below:

C#1: The abstract is overly long, and needs to be curtailed. Further, it should be entirely reorganized and be made precise and to the point summarizing the following:  problem statement, research objectives, data used and methods followed, and specific findings.

C#2: In section 1, the first para authors have introduced the burden of traffic accidents/injuries worldwide as well as in China. However, this should be further expanded. It is recommended to review and include the following important studies in this regard:

·        “Injury severity prediction of traffic crashes with ensemble machine learning techniques: A comparative study”;

·        “Traffic safety in an aging society: Analysis of older pedestrian crashes”; “

·        Adopting machine learning and spatial analysis techniques for driver risk assessment: insights from a case study”;”

·        Explaining self-reported traffic crashes of cyclists: An empirical study based on age and road risky behaviors”;  

·        “A fuzzy-logic approach based on driver decision-making behavior modeling and simulation”;

C#3: Lines 40-51 some safety statistics of E-bicycle users should be included

C#4: Authors should provide justification for selecting and targeting the specific road user groups. Do they have unique crash and safety characteristics, and how they are different from other VRUs groups?

C#5: The introduction section is seriously deficient and brief. It is suggested to comprehensively extend it in light of the latest developments and studies. Further, splitting of introduction into two sections, i.e., introduction and related works is also recommended.

C#6: VRUs is among the critical road user groups and are at the potential in event of crashes.  Recently it has been the subject of growing research attention. This should be clearly emphasized in this literature. For example see and the following studies:  

·        A new pedestrian crossing level of service (PCLOS) method for promoting safe pedestrian crossing in urban areas

·        Evaluating pedestrians' safety on Urban intersections: A visibility analys

·        Enhancing vulnerable road user safety: a survey of existing practices and consideration for using mobile devices for V2X connections

·        Investigation of factors influencing motorcyclist injury severity using random parameters logit model with heterogeneity in means and variances

C#7: Study purpose and specific innovation points should be clearly presented at the end of the introduction section.

C#8: It is further suggested to add a brief description of the paper organization at the end of the introduction section.

C#9: Section 2.1, more information about the study area and reason for selecting the should be provided. Further, how the authors ensured that the sample selected is well representative of study area population using the selected transport mode.

C#10: Why multifactorial logistic regression was preferred for the analysis? Does it suit to the type of data analyzed compared to other contemporary methods? The pros and cons of the methods suggested should be included. Comparison with other advanced statistical analyses will add value to the study.  

C#11: Study limitations and outlook for further studies should be added.  

C#12: Before conclusions, a new para/subsection describing the study's significance from the practitioners' and policy-making viewpoint can be included.   

C#13: While the writinng style and the English language of the paper is generally good, a proofread by a profession native speaker will further improve the text. 

Author Response

Dear reviewer,

Thank you for editor’ and reviewers’ opinions. These comments are very helpful to improve the quality of the manuscript. Now I response the reviewers’ comments with a point by point and highlight the changes in revised manuscript. Full details of files are listed. Thank you again for your valuable comments.

Yours,

sincerely

Point 1: The abstract is overly long, and needs to be curtailed. Further, it should be entirely reorganized and be made precise and to the point summarizing the following:  problem statement, research objectives, data used and methods followed, and specific findings.

Response 1: We appreciate it very much for this good suggestion, and we have revised the abstract according to your ideas.

Point 2: In section 1, the first para authors have introduced the burden of traffic accidents/injuries worldwide as well as in China. However, this should be further expanded. It is recommended to review and include the following important studies in this regard:

  • “Injury severity prediction of traffic crashes with ensemble machine learning techniques: A comparative study”;
  • “Traffic safety in an aging society: Analysis of older pedestrian crashes”; “
  • Adopting machine learning and spatial analysis techniques for driver risk assessment: insights from a case study”;”
  • Explaining self-reported traffic crashes of cyclists: An empirical study based on age and road risky behaviors”;  
  • “A fuzzy-logic approach based on driver decision-making behavior modeling and simulation”;

Response 2: We appreciate it very much for this good suggestion, and we have done it according to your ideas. We have added the burden of traffic accidents/injuries worldwide as well as in China and cited some of your recommended literature, as shown in lines 30-35 and 42-46 of the article.

Point 3: Lines 40-51 some safety statistics of E-bicycle users should be included.

Response 3: We appreciate it very much for this good suggestion, and we have added relevant security statistics according to your ideas, as described in lines 52-62 within the article.

Point 4: Authors should provide justification for selecting and targeting the specific road user groups. Do they have unique crash and safety characteristics, and how they are different from other VRUs groups?

Response 4: We appreciate it very much for this good suggestion, and we have done it according to your ideas. The speed of electric vehicles is faster than that of bicycles and pedestrians, which can have more serious consequences in the event of an accident.

Point 5: The introduction section is seriously deficient and brief. It is suggested to comprehensively extend it in light of the latest developments and studies. Further, splitting of introduction into two sections, i.e., introduction and related works is also recommended.

Response 5: We appreciate it very much for this good suggestion, and we have done it according to your ideas. We have extended the introduction in light of the latest developments and studies. However, the introduction combines many related works for presentation, so we think it is more appropriate not to separate the introduction from the related works.

Point 6: VRUs is among the critical road user groups and are at the potential in event of crashes.  Recently it has been the subject of growing research attention. This should be clearly emphasized in this literature. For example, see and the following studies:  

  • A new pedestrian crossing level of service (PCLOS) method for promoting safe pedestrian crossing in urban areas
  • Evaluating pedestrians' safety on Urban intersections: A visibility analys
  • Enhancing vulnerable road user safety: a survey of existing practices and consideration for using mobile devices for V2X connections
  • Investigation of factors influencing motorcyclist injury severity using random parameters logit model with heterogeneity in means and variances

Response 6: We appreciate it very much for this good suggestion, and we have done it according to your ideas. We have emphasized the VRUs in the literature and cited some of your recommended literature, as shown at lines 34-86.

Point 7: Study purpose and specific innovation points should be clearly presented at the end of the introduction section.

Response 7: We appreciate it very much for this good suggestion, and we have done it according to your ideas. The specific innovation points have been presented at lines 87-90, and the study purpose have been presented at lines 95-98.

Point 8: It is further suggested to add a brief description of the paper organization at the end of the introduction section.

Response 8: We appreciate it very much for this good suggestion, and we have done it according to your ideas. We have added a brief description of the paper organization at lines 93-95.

Point 9: Section 2.1, more information about the study area and reason for selecting the should be provided. Further, how the authors ensured that the sample selected is well representative of study area population using the selected transport mode.

Response 9: We appreciate it very much for this good suggestion, and we have done it according to your ideas. We have provided more information about the study area and reason for selection at lines 103-107 and added the sample size calculation at lines 124-130.

Point 10: Why multifactorial logistic regression was preferred for the analysis? Does it suit to the type of data analyzed compared to other contemporary methods? The pros and cons of the methods suggested should be included. Comparison with other advanced statistical analyses will add value to the study.

Response 10: We appreciate it very much for this good suggestion.Logistic regression is a common method to analyze influencing factors. It's very easy to implement and very efficient to train. Logistic regression is also a good benchmark against which to measure the performance of other more complex algorithms. This is why we adopt logistics regression for analysis. But the disadvantage is that we can't use it to solve nonlinear problems, because its decision surface is linear. Second, it is highly dependent on correct data representation. We have added this to the limitation at lines 320-324. In the future, different statistical analysis methods such as decision tree and K-nearest neighbor can be added to enhance the analytical value of the research.

Point 11: Study limitations and outlook for further studies should be added.  

Response 11: We appreciate it very much for this good suggestion, and we have done it according to your ideas. We have added the study limitations and outlook for further studies at lines 320-324.

Point 12: Before conclusions, a new para/subsection describing the study's significance from the practitioners' and policy-making viewpoint can be included.

Response 12: We appreciate it very much for this good suggestion, and we have done it according to your ideas. We have added a paragraph in lines 296-309 that provides ERTIs prevention from the perspective of practitioners and policy makers.

Point 13: While the writinng style and the English language of the paper is generally good, a proofread by a profession native speaker will further improve the text.  

Response 13: We appreciate it very much for this good suggestion, and we have done it according to your ideas.

Reviewer 2 Report

The paper is very interesting, however it is not well structured.

The introduction lacks a literature review (see https://doi.org/10.1016/j.aap.2023.106996).

The observational study consisted in six intersections with heavy traffic, high pedestrian flow, and clear visibility and traffic signals. However, there are several prior research pointing out the roundabouts as intersections that are difficult to navigate for cyclists and with a significant impact on their safety. Several previous studies analysed the geometry of the roundabout/intersection as potential critical factors (i.e., https://doi.org/10.1016/j.aap.2022.106858) for cyclist crashes whereas it would be interesting if the authors analyse the crash risk at roundabout from the cyclist perspective, investigating their riding behaviour characteristics at roundabouts.

Author Response

Dear reviewer,

Thank you for editor’ and reviewers’ opinions. These comments are very helpful to improve the quality of the manuscript. Now I response the reviewers’ comments with a point by point and highlight the changes in revised manuscript. Full details of files are listed. Thank you again for your valuable comments.

Yours,

sincerely

Point 1: The introduction lacks a literature review (see https://doi.org/10.1016/j.aap.2023.106996).

Response 1: We appreciate it very much for this good suggestion, and we have done it according to your ideas. We have seen and cited https://doi.org/10.1016/j.aap.2023.106996 and added a literature review in the introduction according to your ideas at lines 47-86.

Point 2: The observational study consisted in six intersections with heavy traffic, high pedestrian flow, and clear visibility and traffic signals. However, there are several prior research pointing out the roundabouts as intersections that are difficult to navigate for cyclists and with a significant impact on their safety. Several previous studies analysed the geometry of the roundabout/intersection as potential critical factors (i.e., https://doi.org/10.1016/j.aap.2022.106858) for cyclist crashes whereas it would be interesting if the authors analyse the crash risk at roundabout from the cyclist perspective, investigating their riding behaviour characteristics at roundabouts.

Response 2: We appreciate it very much for this good suggestion. It is interesting to investigate the behavioral characteristics of EB riders at roundabouts, but we did not conduct the survey at the roundabout because of the safety of the observer and the observation perspective. If we have the opportunity, we will conduct a study at the roundabout through video and other forms.

Reviewer 3 Report

The manuscript describes the results from a numerical analysis of e-bikers' behavioral patterns in the city of  Shantou, China

The methodology is well-known and appropriate, results explained but the study approach is general. What is the research question behind? Just showcasing what happens in Shantou? In that case, why Shnatou is so relevant? How the results achieved may advance knowledge? How they can be useful elsewhere?

Moreover, the manuscript goal is to focus on behaviors, but accidents and in general safety problems are due to more intercating factors which might affect behaviors as typically types of vehicle and quality of maintenance, traffic conditions, weather, time, etc.. None of the above is mentioned. Please report the reasons why or integrate the results accordingly, as this seems a big weakness of the manuscript.

For the non-Chinese readership, can you describe (with pictures, possibly) the local riding/street environment? Especially the ones around the schools  

The discussion section looks like a proxy for an extended summary. Discussion should stress implications and assessments, not just resume results; in the end, there is no need to write a manuscript to conclude that to reduce the occurrence of  dangerous riding behavior traffic police enforcement is needed...this is well-known. this part is not fit in the present form, please revise totally

minor issues

r.117 -118 avoid commercial names/brands; state what that given software processes, instead

r.164 -"the longer the one-way distance travelled": what are typica distances? how far is "longer"?

In the present form the manuscript is just a narrative of the numerical data process without a real contribution to the literature. Please focus on the discussion session, improve the quality of the data interpretation (consequences/implications at policy and regulatory levels, educational issues, healthcare costs, for examples...), also by improvig the supporting literature review and resubmit

Author Response

Dear reviewer,

Thank you for editor’ and reviewers’ opinions. These comments are very helpful to improve the quality of the manuscript. Now I response the reviewers’ comments with a point by point and highlight the changes in revised manuscript. Full details of files are listed. Thank you again for your valuable comments.

Yours,

sincerely

Point 1: The methodology is well-known and appropriate, results explained but the study approach is general. What is the research question behind? Just showcasing what happens in Shantou? In that case, why Shantou is so relevant? How the results achieved may advance knowledge? How they can be useful elsewhere?

Response 1: We appreciate it very much for this good suggestion. The research aimed to determine the cycling behavior characteristics of cyclists and to explore the risk factors for electric bicycle road traffic injuries (ERTIs). The results may provide a scientific basis for the prevention and control of ERTIs. But this study was conducted solely in Shantou, a small to medium-sized city located in the coastal monsoon area of southeast Guangdong, China, and parents of young children and elementary school students mostly rely on e-bikes for transportation, while other students with independent mobility tend to use e-bikes instead of walking. So the findings cannot be generalized to the entire country or the entire world, and we have added this to the limitations at lines 326-329.

Point 2: Moreover, the manuscript goal is to focus on behaviors, but accidents and in general safety problems are due to more intercating factors which might affect behaviors as typically types of vehicle and quality of maintenance, traffic conditions, weather, time, etc.. None of the above is mentioned. Please report the reasons why or integrate the results accordingly, as this seems a big weakness of the manuscript

Response 2: We appreciate it very much for this good suggestion. We are very sorry for our lack of consideration. We focused on cyclists' safety perceptions and riding behaviors while neglecting some factors that may influence those behaviors, such as traffic conditions, weather, etc. So we have added this to the limitations at lines 324-326.

Point 3: For the non-Chinese readership, can you describe (with pictures, possibly) the local riding/street environment? Especially the ones around the schools

Response 3: We appreciate it very much for this good suggestion, and we have done it according to your ideas. We have described the local riding conditions and street environment at lines 103-106 and 326-328. We didn't think it was appropriate to place the photo in the article. But as you can see in the picture below, similar scenarios are common in Shantou City.

Point 4: The discussion section looks like a proxy for an extended summary. Discussion should stress implications and assessments, not just resume results; in the end, there is no need to write a manuscript to conclude that to reduce the occurrence of  dangerous riding behavior traffic police enforcement is needed...this is well-known. this part is not fit in the present form, please revise totally

Response 4: We appreciate it very much for this good suggestion, and we have done it according to your ideas. We have added a paragraph in lines 296-309 that provides the measures for ERTIs prevention and control in light of the current prevalence of ERTIs.

Point 5: r.117 -118 avoid commercial names/brands; state what that given software processes, instead

Response 5: We appreciate it very much for this good suggestion, and we have done it according to your ideas. We have stated what that given software processes in lines 134-137.

Point 6: r.164 -"the longer the one-way distance travelled": what are typica distances? how far is "longer"?

Response 6: We appreciate it very much for this good suggestion, and we found that it was inappropriate and deleted it.

Point 7: In the present form the manuscript is just a narrative of the numerical data process without a real contribution to the literature. Please focus on the discussion session, improve the quality of the data interpretation (consequences/implications at policy and regulatory levels, educational issues, healthcare costs, for examples...), also by improvig the supporting literature review and resubmit

Response 7: We appreciate it very much for this good suggestion, and we have done it according to your ideas. We have added a paragraph in lines 296-309 that provides the measures for ERTIs prevention and control in light of the current prevalence of ERTIs.

Round 2

Reviewer 1 Report

I appreciate the efforts made by the authors in revising the article thoroughly. I believe most of previous comments were addressed satisfactorily. Therefore, I am happy to recommend it for publishing. 

Author Response

Thank you so much for your comments, affirmation, support, and help with the research of ijerph-228021.

Reviewer 2 Report

Thank you for addressing my concerns. Since you analysed the safety of riders, I think that at least in the limitations of the study you should add a clause on why you did not investigate the roundabouts even if they are considered intersections that are difficult to navigate for cyclists and with a significant impact on their safety (i.e., https://doi.org/10.1016/j.aap.2022.106858) and also add future directions of your study otherwise it seems that your study focused on a topic that it is actually not the core issue for cyclists' safety.

Author Response

Thank you so much for your valuable comments, affirmation, support, and help with the research of ijerph-228021. We have revised according to your ideas. We have added the limitation and future direction and cited your recommended literature, as shown in lines 319–323 of the article. Thank you again for taking time out of your busy schedule to review our article. Best wishes to you!

Reviewer 3 Report

Although the authors have met some revision requirements, still the manuscript is just a narrative of a case study and no specific scientific advances are evidenced. All the elements of weakness from the previous version remain

Unfit for publication

Author Response

Thank you so much for your comments, affirmation, support, and help with the research of ijerph-228021. It is very regrettable that our last resubmission did not meet your expectations. This study highlighted the need to focus on middle school students under 16 years of age to reduce the occurrence of risky riding behaviors through increased safety education and more effective management systems. The elements of weakness have been tried to elucidate in the limitations. Thank you again for taking time out of your busy schedule to review our article. Best wishes to you!